# Long-Term Follow Up of Patients Treated for Inflammatory Bowel Disease and Cytomegalovirus Colitis

**DOI:** 10.3390/diagnostics14182030

**Published:** 2024-09-13

**Authors:** Gurtej Singh, Clarissa Rentsch, William Beattie, Britt Christensen, Finlay Macrae, Jonathan P. Segal

**Affiliations:** 1Department of Gastroenterology, Royal Melbourne Hospital, Parkville 3052, Australia; 2Department of Gastroenterology, University Hospital Geelong, Geelong 3220, Australia; 3Department of Medicine, The University of Melbourne, Parkville 3010, Australia

**Keywords:** inflammatory bowel disease, cytomegalovirus, colitis, ulcerative colitis, Crohn’s disease, recurrence, colectomy

## Abstract

Background: Pathological reactivation of latent Cytomegalovirus (CMV) is triggered by inflammation and immunosuppression; both present in the pathogenesis and treatment of Inflammatory Bowel Disease (IBD). Whether CMV reactivation is associated with escalating medical therapy, further hospital admissions, or worse clinical outcomes remains controversial. This study aimed to follow up IBD patients with an index episode of CMV colitis and analyse the clinical outcomes. Methods: A retrospective study of patients with IBD treated for CMV colitis was completed. The outcome results were collected at 6-month and 12-month time points after the first episode of CMV colitis. A total of 13 patients with Ulcerative Colitis and 1 with Crohn’s Disease were included. Results: CMV colitis recurrence occurred in 29% of patients at 12 months. A total of 43% of patients had changed their biologic dose at 6 months and 29% had escalated their biologic dose at 12 months. At 12 months, 36% of patients had been re-hospitalised, including three colectomies. Disease remission was only achieved by 29% of patients at 12 months. Conclusions: IBD patients with CMV colitis have substantial rates of re-hospitalisation, failed medical therapy, and colectomy. These risks may be greater at <6 months from an index episode of CMV colitis.

## 1. Introduction

Cytomegalovirus (CMV) is from the Herpesviridae family. Infection is common in humans with seroprevalence rates existing between 40 and 100% [1]. CMV is transmitted via exchanges in secretions and biological fluids such as saliva, milk, semen, and urine [2]. Primary CMV infections tend to be either asymptomatic or cause mild and self-limiting illnesses [3], but a viraemia allows the virus to spread to organs throughout the body. Following primary infection, CMV persists as a latent life-long infection in several cell types including myeloid progenitors, monocytes, and endothelial cells; all of which exist in large numbers in the colon [4,5].

Immunodeficiency remains the primary risk factor for reactivation of CMV from its latent state. CMV reactivation can result in end-organ disease including colitis, pneumonia, hepatitis, retinitis, and gastritis. In post-liver transplant patients, the total incidence of all CMV end-organ diseases at 10 years post-transplant was 4.9% [6]. Patients post allogenic haematopoietic stem cell transplant have an incidence of CMV end-organ disease of 15–25% [7]. Immunocompetent hosts rarely experience CMV reactivation unless they are critically unwell or heavily comorbid. Meta-analysis has shown that 36% of critically ill immunocompetent patients in ICU had CMV end-organ disease [8]. The percentage of immunocompetent patients who suffer from CMV disease is estimated to range between 64 and 75 and is associated with underlying comorbidities such as cardiomyopathy, diabetes mellitus, and chronic kidney disease [9,10].

CMV colitis refers to CMV reactivation specifically in the colon of patients. As mentioned previously, immunosuppression is the fundamental risk factor for CMV colitis. Adults with HIV, malignancy, and solid organ transplants have been shown to experience CMV colitis, and children with prolonged lymphopenia secondary to leukaemia are associated with CMV disease, including CMV colitis [11]. 

Outside of typical immunodeficiency diseases, CMV colitis is also seen in Inflammatory Bowel Disease (IBD). IBD encompasses two chronic intestinal disorders: Crohn’s Disease (CD) and Ulcerative Colitis (UC), which are characterised by immune-mediated inflammation of the gastrointestinal tract. CD pathogenesis consists of a transmural inflammation that occurs in non-contiguous lesions, which may be distributed anywhere in the alimentary tract from the mouth to the anus. UC is characterised by a superficial mucosal inflammation that extends in a contiguous manner proximally from the rectum and is limited to the colon [12]. The disease history of IBD typically comprises alternating periods of acute inflammatory flares and periods of asymptomatic remission. Treatment options for IBD broadly aim to first induce remission then achieve maintenance. This is achieved through anti-inflammatory therapies such as steroids, immunomodulators, and more recently, biologics. The symptoms of CMV colitis typically mirror the full spectrum of IBD symptoms from diarrhoea and per-rectal bleeding to weight loss and colonic perforation [9,13].

The prevalence of CMV colitis in total IBD flares is estimated to be 1.5–4.5% [14,15]. It is more common in UC patients than those with CD with a prevalence of up to 40% in UC patients during acute severe flares [16,17,18]. The prevalence of CMV reactivation in moderate-to-severe IBD flare-ups ranges from 21 to 34% [19,20] and is greater than 30% in steroid-refractory patients [1,21].

Within the IBD population, risk factors for CMV colitis include older age [17], shorter disease history [22], more severe disease seen on endoscopy [16], and pancolitis [23]. Immunosuppression therapy used in IBD management also increases the risk of CMV colitis. In patients with active UC, high doses of steroids are an independent risk factor for CMV colitis [16,23]. Other commonly utilised immunomodulators in IBD such as thiopurines and methotrexate are also associated with CMV colitis, but TNF-inhibitors are not [24,25].

The inflammatory processes inherent to IBD are also purported to predispose patients to CMV colitis. CMV is not an inflammatory virus itself and it is thought reactivation occurs through inflammation and damage to the intestinal mucosa. Once reactivated, it can perpetuate flares, mucosal damage, and shorten remission periods [2]. This being said, it remains unclear whether CMV infection is an innocent bystander in inflammatory intestinal processes or if it increases the risk of adverse outcomes, such as escalating medical therapy, re-hospitalisation, or colectomy [26,27,28]. Some studies have suggested higher CMV loads recognised with histopathology to be associated with higher colectomy rates [29,30], while others have suggested CMV infection does not cause increased morbidity or mortality in patients with UC flares [27]. As such, supporting evidence is required describing clinical outcomes in IBD patients who have CMV colitis to help prognosticate disease progression and guide therapeutic decisions.

The aim of our study was to describe clinical outcomes for IBD patients treated for confirmed CMV colitis at our single centre. In particular, the described outcomes were as follows: (1) requirement for escalated IBD therapies; (2) further hospitalisations due to IBD or CMV colitis; and (3) requirement of colectomy.

## 2. Materials and Methods

### 2.1. Study Population and Design

A retrospective study of patients treated at Royal Melbourne Hospital, Parkville, Australia was completed. The IBD pharmacy and pathology databases were manually searched from January 2018 to December 2023. The pathology database contained all histopathologic reports including gross and microscopic observations in biopsies from any type of procedure. We first limited our search to contain samples from endoscopy and then searched the database for the terms ‘cytomegalovirus’ and ‘CMV’. The IBD pharmacy databases contained all medical therapies delivered to patients with IBD and their respective indications. We searched this database for patients treated with Ganciclovir or Valganciclovir under the indication of CMV colitis. Any patients deemed eligible for the study from the above searches had their medical records comprehensively reviewed for inclusion. Medical records were obtained using Epic Electronic Health Records (Wisconsin, United States).

Patients were included if (1) they had a known diagnosis of IBD (diagnosed using standard clinical, endoscopic and histological parameters), (2) presented with an acute colitis flare, and (3) were found to be CMV positive through immunohistochemistry or CMV inclusion bodies seen in gastrointestinal biopsies. Patients were excluded if their gastrointestinal immunohistopathology was not positive for CMV, if CMV infection did not represent colitis (e.g., CMV oesophagitis), if the episode of CMV colitis was not the first episode of CMV colitis in that patient’s history (i.e., a recurrence), if CMV infection was associated with other causes of immunosuppression (e.g., transplantation), or if data were incomplete.

### 2.2. Clinical Measures and Outcomes

For patients included in the study, demographic details of age, sex, type of IBD diagnosis, smoking status and duration of IBD diagnosis were collected. Laboratory investigations of C-Reactive Protein [CRP], Albumin, and Faecal Calprotectin at time of first episode of CMV colitis were also collected. Data regarding treatment for IBD including steroid, immunomodulator, and biologic therapy at time of CMV colitis were also collected. Comorbidity was measured via the Charlson Comorbidity Index, which was calculated based on medical history at time of index CMV colitis [31].

The outcome measures collected included the following: a subsequent course of CMV treatment (defined as requiring a subsequent dose of anti-viral therapy after a period of symptom resolvement), a subsequent course of corticosteroids, an escalation in biologic dose, change in biologic therapy, further hospitalisation related to CMV colitis or IBD (defined as ≥48 h stay in hospital), a colectomy, and remission (defined as clinical remission and histopathological or sonographic evidence of quiescent disease). The outcome measures were collected at the 6-month and 12-month time points after an index episode of CMV colitis.

### 2.3. Statistical Analysis

Due to the uncommon occurrence of CMV colitis in this population, we anticipated small patient numbers in this study that would be inadequate to perform robust statistical analysis. As such, analysis was limited to descriptive statistics. Statistics of mean, median, standard deviation, and interquartile ranges were reported for continuous measures. Proportions of outcomes were analysed as percentages. Descriptive statistical analysis was performed using Microsoft Excel (Microsoft Corporation, Redmond, WA, USA, 2024).

### 2.4. Ethical Considerations

The Royal Melbourne Hospital Office for Research Ethics & Governance team approved this study (project number QA2023095).

## 3. Results

### 3.1. Patients

The pathology database search generated 2958 histopathology reports. Of these, 28 reports included some description of CMV positivity in patients with IBD having a colitis episode. The pharmacology database search generated 33 patients. Combining these, a total of 61 patients were identified as eligible for this study. Of the 28 patients identified from the pathology database, 26 were excluded due to incomplete data. Of the 33 patients identified from the pharmacy database, 15 were excluded due to the episode being a recurrence of CMV colitis, 2 due to absence of colitis symptoms, 3 due to being treated for CMV oesophagitis, and 1 due to incomplete data. The remaining 14 patients were included in the final analysis (Figure 1).

The summary of included patient’s demographic and clinical characteristics are listed in Table 1. A total of 13 of the included patients had a diagnosis of UC versus 1 with CD. Patients were on a range of IBD maintenance therapies with mesalazine being the most common (*n* = 7). Seven patients were being treated with biologics. The Charleston Comorbidity index average across all patients was 0.57.

### 3.2. Clinical Measures and Outcomes

The clinical measures at time of CMV colitis are listed in Table 2. Eight patients were treated with IV anti-viral therapy and six were treated with oral therapy. Twelve patients had a concurrent course of corticosteroids.

Clinical outcomes at 6 and 12 months are listed in Table 3. Recurrence of CMV colitis occurred in 21% of cases at 6 months, increasing to 29% at 12 months. Additional courses of corticosteroids remained constant at 21% of cases at 6 and 12 months. A total of 43% of patients had changed their biologic dose at 6 months, and this stayed constant at 12 months. Similarly, 29% of patients had escalated their biologic dose at 6 months which was the same at 12 months. At 6 months, 43% of patients had been re-hospitalised due to IBD or CMV including two colectomies. At 12 months, these figures increased to 36% and three, respectively. Disease remission was only achieved by 7% of patients at 6 months, increasing to 29% at 12 months.

## 4. Discussion

Our retrospective study showed descriptive outcomes at 6 and 12 months in IBD patients who experienced at least one episode of CMV colitis. In our study, 29% of patients would go on to have another episode of CMV colitis by 12 months while 21% would be treated with corticosteroids for another flare-up in this time. Biologic dose modification was present following CMV colitis, with 29% escalating the dose of their current biologic, while 43% changed to a different biologic by 12 months. In a 10 year follow up, Oh, Lee [32] showed a CMV colitis recurrence rate of 57% in 36 patients with UC, with 75% of recurrences occurring within 8 months of an index episode, which is greater than our 29% recurrence rate at 12 months. However, only 14/36 patients in Oh et al.’s study received anti-virals compared to our study where 14/14 received anti-viral therapy. This infers that anti-virals may play an important role in reducing the risk of recurrence. Additional to this, our study featured newer biologic treatments which may be superior in reducing chronic inflammation, and thus may reduce the risk of reactivation. The recurrence rates of 21% at 6 months and 29% at 12 months are still substantial and may suggest that anti-viral therapy should be continued for longer durations, such as 6 months, to limit recurrences. This must be weighed against the known side effects of ganciclovir and valganciclovir such as myelotoxicity, nephrotoxicity, and hepatotoxicity [33]. It is also not clear whether recurrence rates would be similar after cessation of anti-viral therapy and further research is needed to assess this. Our study also showed that further hospitalisation occurred in 36% of patients by 12 months and 21% of total patients had colectomies by 12 months. Remission was only achieved in 7% of patients at 6 months, improving to 29% at 12 months.

To our knowledge, this is the first paper that has investigated changes in IBD biologic therapy following episodes of CMV colitis. More than half of our patients either escalated or changed their biologic dose at 12 months. This may suggest that CMV may have a further antagonistic property with an underlying IBD that predisposes patients to require additional therapies. These inferences would need to be tested against a control group matched for biologic use and more research into the causative factors of biologic failure is generally needed.

The rate of further hospitalisations in patients with CMV colitis over 10 years has been recorded at 44.7%, including 21.1% having colectomies [32]. Similar colectomy rates have been reported at 18.7% by Zagórowicz, Bugajski [28] over a mean follow up period of 3.4 years. Our colectomy rates were consistent with these studies; however, our hospitalisation rate was slightly lower at 36%. Given that our study featured patients on newer biologics not included in previous studies, this may indicate that newer biologic therapies improve hospitalisation rates but do not change colectomy rates following CMV colitis. Again, this hypothesis would require further testing, with groups of different biologics compared against each other.

This study was limited by small patient numbers, reducing the statistical power of the study and limiting the generalisability of the findings. This is not unexpected given the uncommon occurrence of CMV colitis in the IBD population, with a large single-centre study showing a prevalence rate of 1.37% (14/1023 patients) over an 8-year period [15]. Future studies should aim for longer follow up or multi-centre involvement to increase patient numbers and facilitate robust statistical analysis. Additionally, the retrospective nature of this study limits the ability to control for potential confounding factors and biases such as clinician selection on which patients received anti-viral therapy. This lack of control group in the study makes it difficult to determine the direct impact of CMV versus the inflammatory processes of IBD on the clinical outcomes observed. Despite these limitations, our study suggests that CMV colitis may be associated with adverse clinical outcomes in patients with IBD. Strategies to manage CMV colitis in this population are still needed.

Future research should focus on larger prospective studies with control groups to validate these findings. Studies should aim to include a diverse population of IBD patients and compare outcomes between those undergoing an IBD flare with and those without CMV to better isolate the specific impact of CMV on IBD progression. Additionally, investigating the optimal timing and duration of anti-viral therapy in preventing CMV recurrence could provide more definitive guidance for clinical practice. Research into the mechanisms by which CMV exacerbates IBD and influences biologic therapy effectiveness could uncover new therapeutic targets. Finally, exploring patient outcomes across different biologic agents could help tailor more personalized treatment approaches for IBD patients with CMV colitis, potentially improving the long-term clinical outcomes.

## Figures and Tables

**Figure 1 diagnostics-14-02030-f001:**
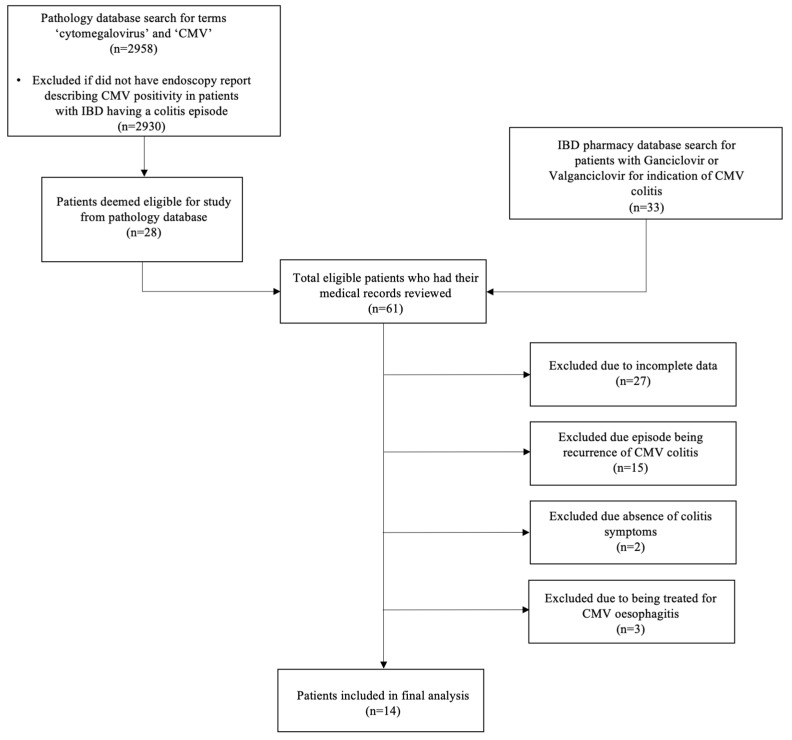
Consort flowchart showing inclusion and exclusion of patients included in final analysis.

**Table 1 diagnostics-14-02030-t001:** Patient and disease demographics.

Measures	Patients Treated for CMV Colitis (*n* = 14)
Sex, *n*	
Female	5
Male	9
Age at episode of CMV colitis, years	
Mean (SD)	55.94 (18.17)
Range	27.0–78.5
IBD type, n	
Ulcerative Colitis	13
Crohn’s Disease	1
Disease Duration at time of CMV colitis	
≤5 years	6
>5 years	8
Smoking status, n	
Current	0
Ex-smoker	4
Never	10
Inflammatory Bowel Medications at time of CMV colitis, n	
Mesalazine	7
Sulfasalazine	2
Azathioprine	3
Mercaptopurine (6 MP)	2
Allopurinol	2
Methotrexate	3
Budesonide	1
Prednisolone	1
Etrolizumab	1
Infliximab	3
Golimumab	1
Vedolizumab	1
Tofacitinib	1
None	4
Charleston Comorbidity Index	
Mean	0.57
Range	0–3

**Table 2 diagnostics-14-02030-t002:** Clinical measures at time of CMV colitis.

Measures (at Time of Diagnosis of CMV Colitis)	Value
CRP, mg/L	
Mean	123.13
Median (IQR)	128 (155.1)
Range	0.5–12
Albumin level, g/L	
Mean	26.38
Median (IQR)	26 (9)
Range	17–37
Faecal Calprotectin, µg/g	
Mean	1792.83
Median (IQR)	1100 (2710)
Range	0–5570
CMV colitis first line treatment	
IV Ganciclovir 5 mg/kg	8
PO Valganciclovir 900 mg BD	6
Concomitant course of corticosteroids during CMV treatment	
Yes	12
No	2

**Table 3 diagnostics-14-02030-t003:** Clinical outcomes in patients at 6 and 12 months after index episode of CMV colitis.

Clinical Outcomes	Patients
At 6 months (percentage of total patients)	
Subsequent course of CMV treatment	3 (21%)
Subsequent course of corticosteroids	3 (21%)
Escalation in biologic dose	4 (29%)
Change in biologic	6 (43%)
Further hospitalisation related to IBD or CMV	6 (43%)
Colectomy	2 (14%)
Remission	1 (7%)
At 12 months (percentage of total patients)	
Subsequent course of CMV treatment	4 (29%)
Subsequent course of corticosteroids	3 (21%)
Escalation in biologic dose	4 (29%)
Change in biologic	6 (43%)
Further hospitalisation related to IBD or CMV	5 (36%)
Colectomy	3 (21%)
Remission	4 (29%)

## Data Availability

The datasets presented in this article are not readily available because of patient privacy and confidentiality. Requests to access the datasets should be directed to G.S.

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
