# Peer review of "Long-Term Follow Up of Patients Treated for Inflammatory Bowel Disease and Cytomegalovirus Colitis"

_diagnostics, 2024, doi:10.3390/diagnostics14182030_

Round 1
Reviewer 1 Report
Comments and Suggestions for Authors
Congratulations to the authors on this significant work.
The methodology applied is appropriate. The results are presented in a suitable manner. The paper cites the appropriate references.
In the discussion, the entire first paragraph represents the authors' results. I would suggest that the authors condense the first paragraph in the discussion and merge it with the second paragraph, where the results are discussed.
This paper is important because it also presents changes in biologic therapy for IBD following episodes of CMV colitis, but the biggest limitation of this study is the small number of participants (which the authors have explained in detail).
Author Response
Comments 1:
Congratulations to the authors on this significant work.
The methodology applied is appropriate. The results are presented in a suitable manner. The paper cites the appropriate references.
In the discussion, the entire first paragraph represents the authors' results. I would suggest that the authors condense the first paragraph in the discussion and merge it with the second paragraph, where the results are discussed.
This paper is important because it also presents changes in biologic therapy for IBD following episodes of CMV colitis, but the biggest limitation of this study is the small number of participants (which the authors have explained in detail).
Response 1: Thank you to the reviewer for reviewing our article and the favourable appraisal. We have condensed the first two paragraphs of the discussion to relate the findings of this study to the literature as suggested by the reviewer (line 186-206).
Reviewer 2 Report
Comments and Suggestions for Authors
General comments to the Authors
An interesting topic. A major revision is necessary.
Major comments
1. The statistical section of this article did not use any statistical method, only using Microsoft Excel for a simple descriptive statistical analysis. Besides, the authors did not set up the control group. I think that the authors cannot get the conclusion “CMV colitis is a negative prognostic marker in patients with IBD, resulting in higher rates of hospitalisation, failed medical therapy, and colectomy. These risks may be greater at <6 months from index episode of CMV colitis”.
2. The authors should draw a flow chart of the exclusion criteria.
Minor comments
TITLE
In the title “Long-term Follow Up of Patients with Inflammatory Bowel Disease treated for Cytomegalovirus Colitis”, I did not think that these patients are treated for CMV colitis alone, but IBD and CMV colitis.
ABSTRACT
In the Background section, the authors said “This study aimed to follow-up IBD patients following an index episode of CMV colitis and analyse clinical outcomes”. The sentence is wrong in grammar. It should be revised as “This study aimed to follow up IBD patients with following an index episode of CMV colitis and analyse clinical outcomes”.
INTRODUCTION
In the sentence (line 59 on page 2) “Ulcerative Colitis is characterised by ……”, the “Ulcerative Colitis” should be revised as “UC”.
Materials and Methods
In the sentence “Patients were included if (1) if they had …”, the word “if” is duplicated.
Comments on the Quality of English LanguageSee my comments above.
Author Response
Comment 1: The statistical section of this article did not use any statistical method, only using MicrosoftExcel for a simple descriptive statistical analysis. Besides, the authors did not set up the control group. I think that the authors cannot get the conclusion “CMV colitis is a negative prognostic marker in patients with IBD, resulting in higher rates of hospitalisation, failed medical therapy, and colectomy. These risks may be greater at <6 months from index episode of CMV colitis”.
Response 1: Thank you for this comment. We agree with your appraisal and have changed the sentence to be "Conclusions: IBD patients with CMV colitis have substantial rates of re-hospitalisation, failed medical therapy, and colectomy. These risks may be greater at <6 months from index episode of CMV colitis."
Comment 2: The authors should draw a flow chart of the exclusion criteria.
Response 2: Thank you for this comment. We have included a consort inclusion / exclusion diagram to illustrate patient inclusion and exclusion.
Comment 3: In the title “Long-term Follow Up of Patients with Inflammatory Bowel Disease treated for Cytomegalovirus Colitis”, I did not think that these patients are treated for CMV colitis alone, but IBD and CMV colitis.
Response 3: Thank you for this comment. We have changed the title to reflect treatment of both IBD and CMV colitis - "Long-term Follow Up of Patients treated for Inflammatory Bowel Disease and Cytomegalovirus Colitis"
Comment 4: In the Background section, the authors said “This study aimed to follow-up IBD patients following an index episode of CMV colitis and analyse clinical outcomes”. The sentence is wrong in grammar. It should be revised as “This study aimed to follow up IBD patients with following an index episode of CMV colitis and analyse clinical outcomes”.
Response 4: Thank you for picking up this error. We have changed the sentence to be "This study aimed to follow up IBD patients with an index episode of CMV colitis and analyse clinical outcomes
Comment 5: In the sentence (line 59 on page 2) “Ulcerative Colitis is characterised by ……”, the “Ulcerative Colitis” should be revised as “UC”.
Response 5: Thank you for this comment. We have changed Ulcerative Colitis to UC.
Comment 6: In the sentence “Patients were included if (1) if they had …”, the word “if” is duplicated.
Response 6: Thank you for this comment. We have deleted the duplicated "if".
Round 2
Reviewer 1 Report
Comments and Suggestions for Authors
Congratulations to the authors on this significant work. The authors have effectively responded to the previous comments, and I believe the manuscript is now suitable for publication in its present form.
Author Response
Comment 1: Dear authors,
Thank you for submitting this interesting manuscript. The reviewers' comments have been appropriately addressed.
In view of the small number of patients included in the study, I would recommend that there are a couple of lines added to this effect. Kindly add in the methodology about the limitations that a small number of patients pose in performing robust statistical analysis. This is in addition to the limitations discussed later in the section. I would also recommend a couple of lines to explain why the numbers are limited- eg uncommon cohort and so on..
Thank you
Response 1: Thank you for your comments - the authors agree with your suggestion. We have added in further detail regarding limited patient numbers in the statsitical analysis section of the methods. We have also added in explanation as to why small patient numbers occurred and suggestions for on how to address these in future studies in the discussion. Please see the changes in the tracked copy of the latest manuscript.